# Behaviour change interventions for the management of Raynaud's phenomenon: a systematic review protocol

Jo Daniels,[1] John D Pauling,[2,3] Christopher Eccelston[4]

► Prepublication history and additional material are available online. To view these files please visit the journal online (http://dx.doi.org/10.1136/bmjopen-2017-017039).

[1]Department of Psychology, The University of Bath, Bath, UK
[2]Department of Rheumatology, Royal National Hospital for Rheumatic Diseases, Bath, UK
[3]Department of Pharmacy and Pharmacology, The University of Bath, Bath, UK
[4]Centre for Pain Research, Department for Health, The University of Bath, Bath, UK

**Correspondence to**
Dr Jo Daniels;
j.daniels@bath.ac.uk

## ABSTRACT

**Introduction** Raynaud's phenomenon (RP) describes excessive peripheral vasospasm to cold exposure and/or emotional stress. RP episodes are associated with digital colour changes, pain and reduced quality of life. Pharmacological interventions are of low to moderate efficacy and often result in adverse effects such as facial flushing and headaches. Recommended lifestyle and behavioural interventions have not been evaluated. The objectives of the proposed systematic review are to assess the comparative safety and efficacy of behaviour change interventions for RP and identify what we can learn to inform future interventions.

**Methods and analysis** Studies eligible for inclusion include randomised controlled trials testing behaviour change interventions with a control comparator. A comprehensive search strategy will include peer review and grey literature up until 30 April 2017. Search databases will include Medline, Embase, PsychINFO and Cochrane. Initial sifting, eligibility, data extraction, risk of bias and quality assessment will be subject to review by two independent reviewers with a third reviewer resolving discrepancies. Risk of bias assessment will be performed using Cochrane risk of a bias assessment tool with quality of evidence assessed using Grading of Recommendations Assessment, Development and Evaluation(GRADE). A meta-analysis will be performed if there are sufficient data. Two subgroup analyses are planned: primary versus secondary RP outcomes; comparison of theoretically informed interventions with pragmatic interventions.

**Ethics and dissemination** This review does not require ethical approval as it will summarise published studies with non-identifiable data. This protocol complies with the Preferred Reporting Items for Systematic Review and Meta-Analysis Protocols (PRISMA-P) guidelines. Findings will be disseminated in peer-reviewed articles and reported according to PRISMA. This review will make a significant contribution to the management of RP where no review of behaviour-change interventions currently exist. The synopsis and protocol for the proposed systematic review is registered in the International Prospective Register of Systematic Reviews (registration number CRD42017049643).

## INTRODUCTION

Peripheral vasoconstriction of thermoregulatory precapillary arterioles and

---

**Strengths and limitations of this study**

► This study will offer the first review of behaviour-change interventions for the treatment of Raynaud's phenomenon (RP), making a significant contribution to our current understanding of treatment options and interventions for this common medical complaint.

► Risk of bias assessment will be performed using the Cochrane risk of bias assessment tool with quality of evidence assessed using GRADE. We will also assess for quality of each psychotherapeutic intervention, further comparing theoretically underpinned and pragmatic interventions.

► We will restrict our review to randomised controlled trials only, which may exclude studies of potential interest. However, we aim to evaluate efficacy of high-quality interventions and use the outcomes of this review to inform future high-quality interventions for RP, therefore we believe this exclusion is appropriate. We intend to discuss noteworthy non-randomised trials or observational studies, however this will not be part of our specified search.

---

arteriovenous anastomoses is a normal physiological response to cold exposure designed to preserve normal core temperature.[1] Raynaud's phenomenon (RP) describes excessive peripheral vasospasm to cold exposure and/or emotional stress. Attacks of RP are associated with characteristic digital colour changes (reflecting blood oxygenation and tissue perfusion). Tissue ischaemia (and subsequent reperfusion) during attacks of RP results in pain and paraesthesia causing distress, loss of hand function and reduced quality of life. RP is common, affecting approximately 10% of people.[2] The majority of patients with RP have a functional vasospastic disorder which, while intrusive, is otherwise benign in nature (termed primary RP). Digital perfusion is generally normal in between attacks. In other circumstances, symptoms of RP occur

as a result of disturbed digital tissue perfusion related to separate underlying pathology (termed secondary RP). Important causes of secondary RP are the autoimmune rheumatic diseases such as systemic sclerosis (SSc). In SSc, endothelial damage leads to an obliterative microangiopathy, characterised by structural changes to the vessel wall and near permanent tissue ischaemia. SSc, despite being rare with an estimated prevalence of 250/million,[3] is often used as the focus of RP research. Cold exposure is the most frequent precipitating factor, although emotional stress may account for up to a third of RP attacks.[4]

Behavioural approaches to RP management include lifestyle and habit reversal interventions such as avoiding cold exposure, conserving heat loss, smoking cessation, increasing exercise and reducing stress levels.[5] It is recognised that adherence with interventions of this nature can be poor with estimates of 30%–50% of patients demonstrating poor compliance, regardless of condition, expected outcome or setting.[6] Pharmacological interventions to induce vasodilation and prevent excessive vasoconstriction of the digital microvasculature also have a role in RP management but the efficacy of such treatments is modest in terms of frequency and severity of RP attacks.[7] Furthermore, pharmacological intervention to promote vasodilation often results in adverse effects such as facial flushing, headaches, fluid retention, dizziness and palpitations.[8] Despite the perceived importance of non-pharmacological interventions, comparative efficacy, adoption and compliance with lifestyle interventions has not been fully evaluated.[5] A number of behavioural interventions have been tested for RP but the comparative efficacy of a range of interventions within different disease populations (primary and secondary RP) has not previously been the focus of a systematic review. The specific objectives of the proposed systematic review are to assess the comparative safety and efficacy of a range of behavioural interventions for the management and treatment of symptoms associated with primary and secondary RP. The secondary objective is to identify what we can learn from the studies reviewed to inform future behaviour change interventions for RP.

## METHODS AND ANALYSIS
### Reporting of protocol and review registration
This protocol follows PRIMSA-P guidelines[9] for the reporting of a systematic review protocol (online supplementary table 1). The synopsis and protocol for this systematic review is registered in the International Prospective Register of Systematic Reviews (http://www.crd.york.ac.uk/PROSPERO),%20registration number CRD42017049643.[10] Reference was also made to the generic Cochrane protocol for pharmacological interventions for the treatment of RP.[11]

### Study selection
Randomised controlled trials (RCTs) testing one or more active treatment interventions with a control comparator arm will be included. Both individual and cluster randomisation will be included providing cluster sites meet other inclusion criteria. Non-RCTs and other intervention study designs will not be included. Blinding in non-pharmacological interventions is not always possible or relevant. Non-blinded studies will be included.

### Participants
Adults (18 years or older) with a diagnosis of RP (primary or secondary). Mixed samples studies will be included. The search will be limited to studies with humans.

### Interventions
We define behaviour change interventions as interventions which target symptomatic relief of RP through directed or advised change in patient-determined behaviour. We are particularly interested in studies consistent with National Institute for Health and Care Excellence guidelines for RP,[5] however we purposefully include all interventions which are designed to change behaviour to improve symptoms. Behaviour change interventions are described variably and may use the following adjectives, all of which will be used in the search strategy: behavio(u)ral therapy, cognitive therapy, education, psychoeducation, biofeedback, clinical psychology, psychological, psychotherapy, self-management, cognitive behavio(u)r therapy and behavioural medicine.

### Outcome measures
There is no consensus over the domains of measurement or measurement technology.[12]

Assessment of RP is largely reliant on patient-reported outcomes, typically captured using instruments that monitor RP symptoms over 1–2 weeks (such as the Raynaud's Condition Score (RCS) diary).[13] The primary outcome measures chosen for our analyses mirror those adopted in a recent generic systematic review protocol for RP[11] and include: the severity/impact of RP episodes assessed using Visual Analogue Scales (VAS); Likert Scales or the RCS either at a single time point (using a standardised recall period) or as an average daily score (obtained from the RCS diary or equivalent from RP symptom diary); frequency of RP attacks (adopted from the RCS diary or equivalent symptom diary approach) reported as average daily or weekly frequency of RP attacks; duration of RP attacks (adopted from the RCS diary or equivalent symptom diary approach) reported as the average daily duration of RP attacks over 1–2 weeks; pain assessed using a VAS or Likert Scale (reporting intensity of pain during RP attacks) and patient assessment of disability due to RP/interference on daily activities, for example, the Scleroderma Health Assessment Questionnaire (HAQ) RP VAS or equivalent. Adverse events (hospitalisation/death) and withdrawals from study will also be included within our primary outcomes.

Secondary outcomes will include physician global assessment of severity/impact of RP; patient global assessment

of function/disability secondary to RP (eg, the HAQ Score); change in digital ulceration (positive/negative); treatment preference and general improvement (self-reported overall improvement). Outcome data on anxiety and depression will also be collated where available. Most RP clinical trials involve assessments over one or more weeks. Sensitivity analyses shall be undertaken if the analyses include trials with marked differences in duration of treatment/assessment.

### Search strategy

A comprehensive search strategy has been created, which includes searching of grey literature. Four sources of peer-review databases will be used: Medline, Embase, PsychINFO and Cochrane. The search will have no language or publication date restrictions, seeking to translate where necessary. Studies must be fully published at the time of search; early view online publications will be included. A draft search strategy is included in the online supplementary tables 2 and 3. Endnote and Covidence software (covidence.org) will be used for data management.

### Selection of studies

All studies generated by the search will be screened by two review authors for eligibility. Full texts will be retrieved for those which meet eligibility, which will then be rated independently by two review authors using a prespecified data extraction form (online supplementary tables 4 and 5). Bibliographies of included studies will also be searched for inclusion of relevant studies, which will then be subject to independent review of eligibility. Discrepancies will be reviewed, with resolution through discussion and consultation of a third author reviewer. Level of agreement will be calculated.

### Data collection process

All eligible studies will be subject to independent data extraction by two review authors. A data extraction form has been iteratively developed and piloted to extract the following information from included studies: full citation including author and year of publication; retrieval information(date/database); patient population; diagnostic criteria used for RP diagnosis; study design; intervention(s); study setting/county (including language); format of intervention (eg, group/individual/internet); sample size and description; age/gender/ethnicity/ intervention(s); duration of study; number of treatment sessions; primary endpoints; secondary outpoints; frequency, duration and severity of RP attacks; outcome/ results; comments, adverse events.

We will also collect information pertaining to the quality of psychotherapeutic interventions: (1) reference to a theoretical model (2) level of therapist training (3) whether the integrity of the intervention was checked. These criteria are drawn from an authoritative review of empirically supported psychotherapies.[14] We will attempt to retrieve missing data with study authors.

The data extraction form will feature a brief eligibility check box with core inclusion criteria: adults with RP; behaviour change intervention with at least one comparator; RCT; RCS or equivalent. All criteria need to be present in order to demonstrate eligibility.

### Risk of bias

Two review authors will independently assess risk of bias on an outcome and study level using the Cochrane risk of bias assessment tool. Unresolved discrepancies will be reviewed by the third review author. We will assess risk of bias according to the following dimensions: random sequence generation (adequate description and method of participant allocation in accordance with standard randomisation); allocation concealment (adequate concealment of group assignment to prevent selection bias); blinding of participants and personnel (adequacy of measures taken to prevent performance bias and conceal group assignment); blinding outcome assessors (adequacy of measures taken to prevent detection bias and conceal group assignment to outcome assessors); incomplete data (adequacy of the management of missing data and potential implications for bias); selective outcome reporting (reporting bias relating to the consistency between prespecified and reported outcomes); other sources of bias (other concerns not covered elsewhere but may lead to a risk of bias). All eligible studies will be rated as high, low or unclear (risk of bias), on each of these dimensions, culminating in an overall risk of bias (high/low/unclear) in accordance with the *Cochrane Handbook for Systematic Reviews of Interventions.*[15] Funnel plots will be used to assess bias and will be stratified based on bias assessment, if considered useful and appropriate (>10 studies).

### Quality of evidence

We will assess the quality of the evidence, and so our confidence in any estimates of effect using Grading of Recommendations Assessment, Development and Evaluation(GRADE): (in)consistency of effect, imprecision, indirectness and publication bias.[16] If possible we will construct a GRADE summary of findings table, using the seven primary outcomes. Where possible, we will express dichotomous outcomes, e.g. no less than mild pain defined as <30 on a 100-point scale,[17] or 50% reduction in the number of episodes, or minimally clinically important differences (MCID).[18]

### ANALYSIS

We will perform a meta-analysis if there are sufficient data, and it is clinically sensible.

Effect sizes will be calculated at post-test between therapies for each comparison with control (using hedges g) by subtracting average score of active arm from average of control arm and dividing by pooled SD. We will also calculate mean/median differences from baseline to post-treatment for each invention and mean differences of change in measures associated with primary outcomes (where appropriate).

The review outcomes are likely to be reported in both dichotomous and continuous data. For dichotomous data we will report the results as summary of risk ratios and CI levels of 95%, pooling those data with identical outcomes and interventions. For continuous data, standardised mean difference and 95% CI will be used due to likelihood of discrepant scales used in the study. The Cochrane handbook will be used for translation of outcomes of scales.

Heterogeneity will be assessed on the basis of study design, participants, interventions and outcomes and will be assessed using forest plots (using 10% level). Higgins $I^2$ statistic, $X^2$ and visual inspection of forest plots will be used to assess homogeneity with thresholds of 50%–75% for moderate heterogeneity and 75% for significant heterogeneity. We anticipate the use of random-effects meta-analysis due to the likely heterogeneity of participant and outcomes data, however fixed models will be used where appropriate.

In the absence of a meta-analysis, a narrative (descriptive) analysis of primary and secondary outcomes will be provided and will include: (1) aggregated data on mean daily frequency/severity/duration of episodes, patient global assessment of disability and changes in RCS (2) analysis of reported design or intervention flaws in behaviour change interventions (3) analysis of study reported outcomes and/or limitations, for the purposes of informing future behaviour change interventions for Raynaud's Phenomenon.

Due to the lack of consesus over reliable measurement endpoints in RP treatment trials, we will specify MCIDs measures to allow clinically meaningful interpretation of change on standardised measures. More specifically: 15-point difference on RCS, 20% difference on VAS and Likert Scales (based previous estimates in RP[12]), use of Reliable Change Index[19] for other standardised measures.

### Planned subgroup analysis

Two subgroup analyses are planned based on specific clinical hypotheses. First, we will analyse outcomes by diagnosis; studies will be divided (where possible) into primary versus secondary RP. Studies which combine both primary and secondary RP will only separate into two subgroups if data are available to do so. Evidence suggests that primary and secondary RP may have different underlying pathogeneses, with studies reporting clear differences in onset, course and prognosis between the groups.[20] Second, we will analyse outcomes based on the presence or absence of identified psychological model of behaviour change, comparing theoretically informed interventions with pragmatic interventions. Interventions that are empirically supported and evidence-based are more likely to be efficacious than their counterparts, therefore subdivision will allow us to identify whether there is a discrepancy and will allow a more meaningful interpretation of findings. Analysis of subgroups will include the aforementioned test of homogeneity to assess potential group effects,

findings will be interpreted accordingly and further analysis performed where appropriate.

**Acknowledgements** The authors thank Elizabeth Sheils for her assistance with the study and protocol preparation and administration.

**Contributors** All authors, JD, JDP and CE were involved in developing the focus of the systematic review, arriving at the final conception of the proposed systematic review through an iterative process. JD produced the first draft of the protocol and supplementary information, coordinated and assimilated comments from JDP and CE on the protocol, approved the final version for publication and is the guarantor of the manuscript. JDP also critically revised successive drafts of the manuscript, provided expert advice and intellectual input regarding the subject area (Raynaud's phenomenon) and approved the final version for publication. CE critically revised successive drafts of the manuscript, provided expert advice on systematic review methodology and approved the final version for publication.

**Funding** This research received no specific grant or financial support from any funding agency in the public,commercial or not-for-profit sectors.

**Competing interests** None declared.

**Provenance and peer review** Not commissioned; externally peer reviewed.

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
