## [Reviewer comments · BMJ Open]

ARTICLE DETAILS

TITLE (PROVISIONAL)	Behaviour change interventions for the management of Raynaud's phenomenon: a systematic review protocol
AUTHORS	Daniels, Jo; Pauling, John; Eccleston, Christopher

VERSION 1 - REVIEW

REVIEWER	Matthieu Roustit Univ. Grenoble Alpes, France
REVIEW RETURNED	14-Apr-2017

GENERAL COMMENTS	This manuscript describes the protocol for a systematic review of behaviour change interventions in patients with primary and secondary Raynaud's Phenomenon (RP). The topic is very interesting considering the limited efficacy of pharmacological interventions. The methods are robust and comply with high quality standards (PRISMA guidelines and GRADE). Sub-groups analyses are relevant. My only concern is the relevance of publishing a protocol that will also be registered in PROSPERO, with the same level of details. This register has gained important visibility in the past few years and is a reference in the field. Minor remark: Page 8, line 11 : what does the "RCS composite score" refer to? Is it different from the usual RCS? If not, why is it considered both as a primary and secondary outcome?
--

REVIEWER	Valentina Llovet Gutiérrez Pontificia Universidad Católica de Chile
REVIEW RETURNED	18-May-2017

GENERAL COMMENTS	It is a very complete and clear protocol. I strongly recommend the use of Epistemonikos as a source of data as well as the ones already described.
---

VERSION 1 – AUTHOR RESPONSE

REVIEWER 1 COMMENT:

Page 8, line 11 : what does the "RCS composite score" refer to? Is it different from the usual RCS? If not, why is it considered both as a primary and secondary outcome?

RESPONSE:

Thank you for your comments. We agree there is a lack of clarity pertaining to the RCS description in this section and the reviewer has correctly identified inappropriate use the term “composite” in the context of the RCS diary parameters.

The amended paragraph below retains the use of the RCS score (a single item NRS or VAS) assessing Raynaud's severity, however we have offered more detail and clarifications of the use of the RCS (and separate items on the RCS), also withdrawing the word "composite". This mirrors the approach taken by Janet Pope the SR protocol we originally cited (8), but we believe is much clearer.

We have changed the outcome measures section (p5, author page numbering) to reflect this clarification:

Assessment of RP is largely reliant on patient-reported outcomes, typically captured using instruments that monitor RP symptoms over 1-2 weeks (such as the Raynaud's Condition Score [RCS] diary). The primary outcomes measures chosen for our analyses mirror those adopted in a recent systematic review of RP8 and include; the severity/impact of RP episodes assessed using visual analogue scales (VAS), Likert scales, or the RCS (either at a single timepoint using a standardised recall period) or as an average daily score (obtained from the RCS diary or equivalent from RP symptom diary); frequency of RP attacks (adopted from the RCS diary or equivalent symptom diary approach) reported as average daily or weekly frequency of RP attacks; duration of RP attacks (adopted from the RCS diary or equivalent symptom diary approach) reported as the average daily duration of RP attacks over 1-2 weeks; pain assessed using a VAS or Likert scale (reporting intensity of pain during RP attacks) and patient assessment of disability due to RP/interference on daily activities e.g. the Scleroderma Health Assessment Questionnaire (HAQ) Raynaud's phenomenon VAS or equivalent. Adverse events (hospitalization/death) and withdrawals from study will also be included within our primary outcomes.

Secondary outcomes will include physician global assessment of severity/impact of RP; patient global assessment of function/disability secondary to RP (e.g. the HAQ score); change in digital ulceration (positive/negative); treatment preference and general improvement (self-reported overall improvement). Most RP clinical trials involve assessments over 1 or more weeks. Sensitivity analyses shall be undertaken if the analyses include trials with marked differences in durations of treatment/assessment.

REVIEWER COMMENT 2:

I strongly recommend the use of Epistemonikos as a source of data as well as the ones already described.

RESPONSE:

Thank you for your recommendation. We have consulted on this issue, taken expert advice and have decided to retain the current approach for considered reasons. Thank you for your recommendation and we will consider this database in the future.